# Iron Treatment in Patients with Iron Deficiency Before and After Metabolic and Bariatric Surgery: A Narrative Review

**DOI:** 10.3390/nu16193350

**Published:** 2024-10-02

**Authors:** Jila Kaberi-Otarod, Christopher D. Still, G. Craig Wood, Peter N. Benotti

**Affiliations:** 1Department of Nutrition and Weight Management, Geisinger Health System Northeast, Scranton, PA 18503, USA; 2The Center for Obesity and Metabolic Research, Geisinger Obesity Institute, Danville, PA 17821, USA; cstill@geisinger.edu (C.D.S.); cwood@geisinger.edu (G.C.W.); pnbenotti@geisinger.edu (P.N.B.)

**Keywords:** iron deficiency, obesity, bariatric surgery, parenteral iron, iron salts, inflammation, heme iron

## Abstract

Iron is an essential nutrient in living organisms with multiple vital functions. Iron deficiency (ID) can cause long term health consequences beyond iron deficiency anemia (IDA). The high prevalence of ID and its long-term effects in patients with obesity and after metabolic and bariatric surgery (MBS) is recognized. Nevertheless, there is limited knowledge of the optimal route or dose for treatment of patients with obesity and post-MBS, and an evidence-based universal guideline for prevention and treatment of ID in short- and long-term post-MBS (PMBS) is not yet available. ID in the general population is currently treated with oral or parenteral iron, where oral iron treatment is considered the preferred option with parenteral iron as a second-line treatment in case there is intolerance or lack of response to oral iron. In patients with obesity with chronic low-grade inflammation and PMBS patients with altered gut anatomy and function, there are also alterations in the bioavailability and higher risks of side effects of available oral irons. The conclusions of current studies exploring effective treatment of iron deficiency in this population have been inconsistent and further well-planned randomized and prospective studies are needed. This is a narrative review of the literature on the available treatment options and strategies for treatment of ID in PMBS patients to recognize the knowledge gaps and provides topics of future research.

## 1. Introduction

With the global rise in the prevalence of obesity, metabolic and bariatric surgery (MBS) has emerged as the most effective and durable treatment of severe obesity and its related disorders. However, permanent surgical alterations in foregut anatomy and physiology contribute to various nutritional deficiencies [1]. Iron deficiency (ID) is now recognized as one of the most common and consequential deficiencies in post-MBS (PMBS) patients [2,3]. Reviews of long-term outcomes in adults and adolescents who underwent MBS procedures suggest a high prevalence of ID in both adults (12–53%) and adolescents (30–70%) [2,3,4,5,6]. These studies have shown that the risk increases with time after surgery [2,3,7,8].

ID is considered common with a prevalence of 42% among United States adults. There is also clinical evidence from epidemiologic surveys and cohort studies to support an association between obesity and ID [9,10]. However the reported prevalence of ID in patients with obesity and candidates for MBS is inconsistent and depends on the diagnostic criteria and methods used to define ID. Analysis of data from the 2001–2006 National Health and Nutrition Examination Survey (NHANES), a population-based cross-sectional study, reported the prevalence of ID to be 9.3–22.9% in non-pregnant females with a body mass index (BMI) > 30 and 5.5–12.5% in those with a BMI of 18.5–25. The prevalence in candidates for MBS is reported to be between 8.7 and 53% [11]. This is concerning because there is evidence that pre-MBS ID is a risk factor for the development of PMBS ID and iron deficiency anemia (IDA) [2,12,13].

Iron deficiency leads to consequences beyond erythropoiesis [14]. ID in female adolescents soon to be at childbearing age with obesity or PMBS is specifically concerning as it can have deleterious effects on the neurons and brain development of the offspring [14]. These effects can be reversible with iron therapy. Untreated ID during the period of rapid growth and differentiation of the brain can result in long-term structural deficit and neurocognitive abnormalities, which can persist even after iron repletion. While hematinic levels in IDA may be normalized with iron treatment, some studies have suggested that the neurological effects of persistent ID may be irreversible [15,16,17,18]. These findings support other studies that emphasize the importance of treating ID in the absence of anemia [14,15].

ID causes symptoms such as fatigue, hair loss, headaches, and restless leg syndrome as well as cognitive and attention deficit even in the absence of anemia. These symptoms alter quality of life and can mitigate the beneficial effects of bariatric surgery. Iron therapy with oral or IV iron has been shown to improve fatigue due to ID and result in a feeling of wellbeing [19,20,21].

The high prevalence of ID in PMBS and its long-term adverse consequences render the prevention and treatment of ID of utmost importance [2,22]. Currently, there is a lack of universal evidence-based guidelines addressing the type, effective dosage, and frequency of iron supplementation to prevent and treat ID and IDA in PMBS, nor are there clear indications for the use of parenteral iron in this patient population [23,24,25]. The purpose of this review is to summarize the most current data on ID treatment for obesity and MBS, to identify knowledge gaps, and to define further areas of needed research.

## 2. Iron Metabolism and Bioavailability and the Effect of Bariatric Surgery

Iron levels in the body are tightly regulated to ensure adequate availability while preventing potential toxicity from iron excess. Control of iron metabolism is mainly exerted by systemic levels of the hepatic-derived protein hepcidin by controlling absorption from the intestinal mucosa and releasing iron from macrophages and iron stores. Normally, hepcidin levels are regulated by body iron content and erythropoietic demands. In the iron-replete state, hepcidin levels are increased, which limits iron export from iron stores and enterocytes into the circulation by degrading ferroportin, the only known exporter of iron on the target cells. In a state of iron deficit, hepcidin levels decrease, allowing more iron to enter the circulation from enterocytes and other iron stores to meet the demand.

Hepcidin and ferritin are both also acute phase reactants [26]. In chronic diseases associated with inflammation, such as obesity, hepcidin synthesis is enhanced and an increased hepcidin level results in iron sequestration and consequently low iron bioavailability to the cells. This causes functional ID, which over time can be accompanied by true iron deficiency because of long-term decreases in absorption [27,28,29]. The effect of inflammation on iron metabolism, although well recognized, has made the criteria for iron deficiency diagnosis in the presence of inflammation challenging and a topic of current research.

Under normal conditions, absorption of iron from the intestine is limited to 1–2 mg of iron daily, which is adequate to replace iron loss. Absorption of iron predominately occurs in enterocytes in the duodenal portion of the intestine. These enterocytes express membrane-associated ferric reductase duodenal cytochrome b (Dcytb) and divalent metal transporter 1 (DMT1) on the apical brush border membrane and ferroportin on the basolateral membrane. Dcytb is a reductase and DMT1 is the primary transmembrane iron transporter into enterocytes. These proteins are both required for absorption of iron from the intestinal lumen into the enterocytes [30,31]. Then, the iron in the enterocytes is exported into the circulation via ferroportin, the universal exporter of iron. Systemic signals can control the expression of Dcytb, DMT1, and ferroportin to regulate absorption of iron [31]. Intestinal absorption and bioavailability of ingested oral iron can vary from <1% to >50% and is determined by the amount of body iron, food bioavailability, and dietary iron content [26,31,32].

Bioavailability of iron in food is influenced by the type of dietary iron. Dietary iron is either heme iron (more bioavailable and less abundant in the diet) or non–heme iron (less bioavailable but more abundant in the diet) (Table 1). As much as 30–40% of heme iron is absorbed by the enterocytes compared with 1–10% of non–heme iron because heme iron is directly taken up by enterocytes and does not require DMT1 for absorption [32,33]. Iron from food is mostly non–heme iron in the ferric (Fe^+3^) form released from food by gastric acid and it requires reduction to the ferrous (Fe^+2^) form by dietary reducers and Dcytb [32,34].

Intestinal absorption of iron and bioavailability are also affected by the gut microbiota (GM). The crosstalk between the GM and the iron supply is an area of current research. It is known that iron deficiency and excess can both impact the GM composition [33,35,36] and the GM may control iron hemostasis by affecting the immune system and inflammation. As a result of these findings, it is postulated that prebiotic or probiotic supplementation can improve iron absorption. There is some evidence that a diet rich in prebiotics and probiotics may improve iron status by increasing iron bioavailability [37,38]. Several studies conducted to assess the effects of iron-fortified food, prebiotic supplements such as pectin, and probiotic supplements with diet or iron treatments were inconclusive [33,39,40].

In obese and PMBS patients, several steps in the absorption and metabolism of iron can be disrupted, resulting in ID. In patients with obesity, ID is multifactorial, and causes may include decreased intake of iron-rich foods due to poor food choices and/or food insecurity and decreased stomach acid content/production as a result of the treatment of the obesity-related comorbidities, such as gastric acid reflux stemming from the use of acid-suppressing medications. Also, iron loss in females with obesity due to irregular and heavy menstruation and gradual blood loss due to gastritis and hemorrhoids in patients with obesity can exacerbate iron deficiency. Moreover, obesity is recognized as a chronic disease and is commonly associated with low-grade inflammation. It is proposed that diet-induced enlarged adipocytes, especially in visceral tissue, are the points of inflammation that trigger overproduction of proinflammatory cytokines. The inflammatory component of obesity affects iron metabolism in multiple ways, including altering its bioavailability by the upregulation and production of hepcidin and the recently recognized iron regulator Lipocalin 2, which causes decreased absorption of iron from enterocytes and its entrapment in reticuloendothelial cells [41]. In line with that, there is some evidence that diet-induced weight loss in patients with obesity may reduce inflammation and serum hepcidin levels, improve iron homeostasis, and resolve ID [42].

ID in PMBS is multifactorial as well. Decreased food intake, altered food composition, a high prevalence of food intolerance (specifically of meat), decreased acid content of the stomach, and surgical alteration of the foregut with bypass of the duodenum as the main site of iron absorption may all contribute to the decreased bioavailability of iron and the resulting ID [11]. All MBS surgeries can result in ID with a higher prevalence in patients who undergo procedures with a more malabsorptive component (gastric bypass [20–55%], biliopancreatic diversion with or without duodenal switch [8–62%]) compared to restrictive procedures (sleeve gastrectomy [<18%], adjustable gastric band [14%]) [43]. While MBS-induced weight loss and reduced adipose tissue are expected to improve inflammation and iron homeostasis, that may not be enough to negate the effects of decreased iron absorption, decreased intake, and lack of compliance with oral supplements PMBS.

### 2.1. Treatment of ID in PMBS Patients

Several studies have demonstrated a high prevalence of ID PMBS even in patients taking their recommended prophylactic multivitamins, and as a result, additional prophylactic iron treatment has been recommended [44,45,46,47]. The current recommendation is to use oral iron for the treatment of iron deficiency with or without mild anemia. The ASMBS/TOS guideline [43] recommends 45–60 mg of elemental iron cumulatively, including from all vitamin and mineral supplements, for the prevention of ID in premenopausal women who have undergone gastric bypass, sleeve, or biliopancreatic diversion with duodenal switch, and the amount should be increased to 150–200 mg daily to 300 mg 2–3 daily for the treatment of ID. Supplementation with intravenous iron is recommended if there is intolerance or no response to oral iron [43]. However, there is a notable lack of agreement in the current guidelines for the prevention and treatment of ID, and the authors of these guidelines acknowledge that the evidence base for these recommendations is limited [24,25,43,48,49,50].

Several short-term studies have shown improvement of ID with the use of iron salts, but longer-term follow-up has failed to improve ID and resulted in the use of intravenous iron as a second-line treatment. A systematic review of studies on post-bariatric ID prevention and treatment found that bariatric surgery programs apply a wide range of prophylactic iron supplementation from 15–105 mg of elemental iron [23]. The prophylactic strategies were only effective in about half of the studies, and in many of the studies, the iron supplementation was inadequate, likely because of side effects (SEs) and/or patient non-adherence to treatment [23,51,52].

There are a large number of different preparations of iron available with wide variation in elemental iron content and bioavailability [53]. To optimize the absorption and bioavailability, it is recommended that oral iron be taken 1 h before or 2 h after food intake (or on empty stomach in the morning) and to be taken 1–2 h apart from other medication and supplements, specifically antacids, proton-pump inhibitors, calcium, and zinc supplements [32]. Phytates (in cereals) and polyphenols (in tea and coffee, fruits, and vegetables) have the most inhibitory effects on non–heme iron bioavailability, and this can be negated by an absorption enhancer such as ascorbic acid [32,54]. There is some evidence that concomitant use of vitamin C can be beneficial [55] by improving bioavailability, and the addition of vitamin C from food sources and/or a supplement formulation is recommended. In addition to iron supplementation, inclusion of iron-rich foods to supply heme iron can be beneficial as a part of a post-bariatric diet. There is evidence that changes in the non–heme iron content of the diet can rapidly affect enterocyte DMT1 expression and function [56,57]. Some short-term studies in PMBS patients demonstrate compliance with recommended dietary changes as well as supplements with positive effects on iron levels [58,59]. However, longer-term evidence supporting the effectiveness of dietary changes with supplements in the treatment and/or prevention of ID is inconclusive. The positive findings reported in the short-term studies may be the result of increased use of resources to provide intensive follow-up, monitoring, patient education, and counseling throughout the research period [59,60,61]. The non-adherence to recommended diet and nutritional supplements increases with time after surgery as well [8,62]. There is a need for well-designed studies to assess the long-term effects of diet on iron levels and hematinic values in PMBS patients.

### 2.2. Pharmacotherapy

#### 2.2.1. Oral Iron

Ferrous sulfate is the most widely used iron preparation due to its lower cost and higher bioavailability. However, iron salts generally have a high rate of intolerance (Table 2) [63,64,65,66,67]. Long-term use of oral iron can have adverse effects on gut microbiota, contributing to increased inflammation of the gut mucosa by impacting the gut microbiota composition [35,68,69,70,71]. There are a few studies comparing different oral iron preparations in post-bariatric patients with different outcomes but, overall, no oral iron preparation has been shown to be effective in the prevention of post-bariatric ID in the long term [51,52,72,73,74]. The high prevalence of gastrointestinal SEs and intolerance results in non-adherence and frequent discontinuation of treatment [75,76]. There is evidence that an elevated level of hepcidin after a dose of iron persists for about 48 h [77], and this may result in decreased absorption of subsequent doses and increased excess iron in the gut lumen, which may cause GI intolerance and SEs [35,69,77,78]. To mitigate the risk of GI SEs, smaller doses and/or a less frequent dosing schedule may be beneficial [79]. It is suggested that a daily or every-other-day dose be considered in the general population with ID, but these methods have not been studied in post-bariatric patients [74,77,80] and there is no consensus on the effective dose of elemental iron for the oral treatment of ID in this population.

Novel iron preparations with improved absorption and tolerance may be beneficial for the long-term prevention and treatment of ID [81,82]. Sucrosomial^®^ iron (SI) contains ferric pyrophosphate enclosed in a phospholipid and sucrester matrix coated with tricalcium phosphate and starch for stability. In vitro and animal studies have shown that SI does not raise hepcidin levels and possibly bypasses the conventional iron absorption mechanism by enterocyte uptake via a vesicle-like structure not involving DMT1 and independent of the hepcidin pathway [82]. This offers the potential of better absorption and fewer GI SEs. Studies of the efficacy of SI in different inflammatory disorders, preoperative blood management, and a short-term treatment of PMBS females with ID demonstrated non-inferiority to IV iron treatment [82]. SI can be a valid alternative option to prevent ID and IDA in PMBS patients in the long term, especially in patients with intolerance to iron salt formulations [83], but there is no long-term study of SI in PMBS patients currently.

Ferric maltol (FM) is another novel non-salt iron preparation composed of a complex of ferric iron and tri-maltol, a sugar derivative. This formulation remains in complex form and allows bioavailability at neutral PH levels with more elemental iron as ferric iron to reach the site of absorption on enterocytes, resulting in a better absorption and less toxic effects [84]. The higher bioavailability and reduced toxicity allow for smaller doses and less tissue damage with a better safety profile [81]. FM has been studied in the treatment of ID in inflammatory bowel diseases and chronic kidney disease with an improvement in hematinic values [85,86]. There are no studies addressing the effect of FM in PMBS patients with ID. FM is not available over the counter.

#### 2.2.2. Intravenous Iron

In patients who fail oral iron therapy due to intolerance, lack of response, or the development of severe ID/IDA with symptoms, treatment with IV iron should be considered. IV iron treatment has been demonstrated to improve hematinic levels faster than oral iron treatment [87,88]. Some researchers increasingly favor the use of IV iron as the first line of treatment for ID in certain conditions such CKD and pregnancy and for PMBS patients because of the effective rapid correction of hematinic levels with fewer adverse effects than oral iron [89]. Hemoglobin levels should increase within 1–2 weeks after treatment, and an increase of 1–2 g/dL within 4–8 weeks of treatment is expected [19,90]. The response to iron treatment should be evaluated in 6–8 weeks to assess for proper repletion and response. Lack of response should be appropriately investigated. However, iron parameters should not be evaluated before 6–8 weeks after the iron infusion since circulating iron interferes with the assays and the results are misleading and cannot be interpreted correctly [19]. The dose of IV iron can be calculated using the Ganzoni formula or a simplified calculation with consideration of weight and hemoglobin level (Box 1) [91].

Box 1Ganzoni formula for the calculation of intravenous iron dose.Total iron dose (mg) = body weight (kg) × (target hemoglobin − actual hemoglobin [g/L]) × 0.24 + iron for iron store ^¥^ (mg)^¥^ Iron stores: body weight < 35 kg = 15 mg/kg body weight and body weight > 35 kg = 500 mg

The previous generation of IV iron preparations, specifically high-molecular-weight iron dextran, were associated with a high risk of severe hypersensitivity reactions, which restricted their use. These formulations have been discontinued due to their unfavorable safety profile and high risk of a serious reaction. The new generation of IV iron supplements has significantly improved safety (Table 3) [92,93]. A systematic review of IV iron preparations demonstrated that IV iron is associated with less risk of GI SEs compared with oral iron salts with no increase in adverse events requiring discontinuation of treatment and no increased risk of infections, which had previously been a major concern. The risk of serious SEs was 1 in 200,000 cases and no anaphylactic reactions occurred [94].

Despite the improvement in safety profile, all IV iron formulations carry the risk of infusion reactions. The associated amount of labile free iron release after an IV infusion can be cytotoxic and results in the Fishbane reaction, which is characterized by arthralgia, chest tightness, and flushing [68,95,96]. Fishbane reaction symptoms are usually mild and resolve shortly after stopping the infusion. The infusion can be restarted with a slower rate after 15 min of symptom resolution and symptoms are unlikely to recur. IV formulations with higher labile iron release (iron sucrose [IS] and ferric gluconate [FG]) should be given in the recommended smaller doses in separate infusions to avoid infusion reactions. Newer preparations such as iron carboxymaltose (FCM), ferumoxytol (FER), iron isomaltoside (FDI), and low-molecular-weight iron dextran contain lower levels of labile iron and can be given as a single-dose infusion (Table 3) [96].

The Fishbane reaction is different from a hypersensitivity reaction as it does not cause angioedema, wheezing, or hypotension [68], and is considered a pseudo-allergy. Free labile iron can activate the complement system, leading to a histamine-producing inflammatory cell response (mast cells and basophils) that mimics an IgE-mediated anaphylactic reaction, which is called a complement activation-related pseudo-allergy (CARPA) [68,95]. The appropriate treatment of infusion reactions demands a trained staff familiar with treatment algorithms [68,97,98,99] to watch for symptoms in a controlled environment for 15–30 min after infusion before discharge. For moderate and more severe cases with hypotension, IV hydration may be required and advanced cardiac support should be available in the unlikely event of cardiac arrest. Additionally, venous canulation/access is needed for IV iron infusion and extravasation, and problems with access should be monitored and treated in the facility.

Hypophosphatemia is increasingly recognized as an SE of current IV iron formulations. It is diagnosed at a phosphorus level < 2.5 mg/dL and considered moderate at <2.5–2 mg/dL and severe at <2–1 mg/dL. A phosphorous level <1 mg/dL can cause severe symptoms and is potentially life-threatening. Several head-to-head studies and a meta-analysis of the studies addressing IV iron formulations all demonstrated a higher incidence of hypophosphatemia consistently associated with FCM and, to a lesser extent, FDI or FER [100,101,102]. The mechanism for the development of hypophosphatemia is thought to be related to the effect of IV iron on intact fibroblast growth factor 23 (iFGF23), a phosphaturic hormone. FCM triggers a cascade of events with an increased level of iFGF23 followed by decreased 1,25-dihydroxy vitamin D and calcium, and secondary hyperparathyroidism ensues [103]. The increased parathyroid hormone (PTH) continues the vicious cycle of hypophosphatemia with prolonged elevation of PTH even after iFGF23 normalizes [103]. Hypophosphatemia may occur within the first 2 weeks of infusion, and retrospective studies have shown that it may persist for 6 months after infusion (median time of 40–80 days) [101,102], but there is no prospective study addressing the duration of hypophosphatemia after IV treatments. The symptoms of hypophosphatemia include proximal myalgia and weakness, bone pain, and fatigue, and can be confused with ID/IDA symptoms. The diagnosis of hypophosphatemia requires a strong clinical suspicion. Generally, symptoms are mild and can be observed, and routine monitoring is not recommended. However, clinical suspicion should be raised in high-risk patients, such as PMBS patients with malabsorption, and they should be considered for monitoring of phosphorous levels and signs of hypophosphatemia [91,104]. There are reports of a long-term risk of bone complications, osteomalacia in adults, and growth retardation in younger patients [105,106,107] with persistent hypophosphatemia in patients who require frequent administration of IV iron.

The studies of IV iron treatment for ID in PMBS patients demonstrated that, overall, IV iron is safe and effective in the treatment of ID [108,109,110,111,112,113,114], but the long-term efficacy of IV iron in the prevention of ID and its cost-effectiveness in PMBS remains unknown. In a randomized controlled trial comparing oral and IV iron using a single dose of FCM after gastric bypass surgery, supplementation with a daily dose of oral iron failed to normalize hematinic levels in the majority of the oral iron group at 3 months while all of the study patients in the FCM arm showed improvement with normalization of ferritin at 3 months [111]. At the end of the study at 12 months, approximately 70% of patients in the oral iron group and approximately 30% in the FCM group developed ID/IDA [111]. Although this study is limited by a small sample size and a high dropout rate, the findings suggest that long-term prevention of ID/IDA in PMBS patients will require intermittent iron infusions and oral iron utilized for maintenance. The effective type and dosage of the iron infusion, the optimal frequency of treatments, and the efficacy of oral iron maintenance in PMBS patients are unknown but some retrospective studies suggest regular iron infusions may prevent ID/IDA up to 5 years postoperatively [108,110,115]. All IV iron formulations studied in PMBS patients have been shown to improve hematinic levels with minimal adverse reactions and better tolerance than oral supplements [109,110,111,114]. Therefore, the choice of IV iron formulation in the treatment of PMBS individuals has similar considerations to those in the general population with ID/IDA and should be individualized according to the adverse reaction profile, the number of infusions needed to deliver the total dose requirements, the availability, and the cost. A short-term study of treatment of ID in PMBS patients with SI vs. IV iron showed that SI was as effective as a single dose of IV iron to improve hematinic levels and that it could be an alternative to IV iron, but further well-designed studies are needed to evaluate the long-term effects of SI on post-bariatric iron nutrition [83].

Iron nutrition status has been widely studied in non-bariatric surgeries, and there is some evidence that treatment of ID with or without anemia improves postoperative outcomes [116,117]. There is an expanding list of recommendations [118] for the evaluation and treatment of ID (with or without anemia) prior to non-MBS surgeries but specific guidelines for the diagnosis and treatment of ID varies for different types of surgery [119,120]. ID in patients prior to MBS is common and is linked to the low-grade inflammatory state of the obesity causing functional ID accompanied by true ID resulting from decreased intake and reduced absorption of iron [13,121]. There is emerging evidence that preoperative ID in candidates for bariatric surgery is a risk factor for severe anemia postoperatively in the long term. In the short term, it may affect post-surgical outcomes [2,7,13]. Retrospective studies have shown a possible effect of ID on complications and length of stay [13,122], but there is lack of prospective studies on the effect of treatment of ID on the short-term outcomes of bariatric surgery. It is recommended that preoperative patients undergo a screening for nutritional deficiencies, including iron deficiency, prior to MBS [7,43,123], but currently there is no guideline specific to the treatment of ID in patients with obesity undergoing MBS and studies addressing the effectiveness of preoperative oral or IV iron treatment on the prevention of postoperative ID and outcome are scarce [117].

## 3. Conclusions

The obesity epidemic and the proven efficacy of MBS for improving health and quality of life suggest that the volume of surgery is expected to rise in the future and, as such, an increasing number of patients will need prevention and treatment of surgery-related ID. Unfortunately, despite the increasing evidence pointing to the negative short- and long-term effects and complications of ID/IDA, ID management in severe obesity and in association with bariatric surgery remains an unsolved, but potentially modifiable, nutritional complication.

There is also a higher burden of medical resources utilization and cost [124] in patients with ID/IDA compared with non-anemic patients with normal iron status. Despite these findings, most PMBS patients with ID/IDA are not on oral iron and do not receive IV supplementation [108,124,125]. As such, ID remains an important quality improvement topic for multidisciplinary weight management/MBS programs.

Additional studies on iron supplementation strategies are needed to prevent and treat ID in PMBS patients. A systematic review of randomized controlled trials of bariatric surgery has shown an increased number of studies on MBS over the past decade. However, less than 20% of the studies address nutritional deficiencies [126]. There are only a small number of trials studying the long-term effects of MBS. However, considering the life-long effect of surgery and the chronic nature of obesity, long-term studies are essential.

There are several unanswered questions about ID concerning the appropriate time and frequency of screening, the best diagnostic criteria, the best treatment options, and the proper long-term monitoring strategies to prevent ID/IDA. There is a need for more research opportunities in the prevention and treatment of nutritional deficiencies and long-term effects beyond the obesity-related outcome of surgery to address these knowledge gaps.

## Figures and Tables

**Table 1 nutrients-16-03350-t001:** Examples of Iron Rich Foods.

Iron-Rich Food Sources	Type of Iron	
	Heme Iron Sources	Non–Heme Iron Sources
Grains		Fortified cereals
Protein foods	Oysters, mussels, duck breast, turkey, egg, beef, lamb, shrimp, organ meats, game meats	Sesame seeds, cashews
Vegetables		Spinach, artichokes, soybeans, beans (lima beans), Swiss chard, lentils, beets, mushrooms, leeks, potato with skin
Fruits		Prune juice

Information obtained from an official website of the United State government dietary guidelines for America (DGA) at https://www.dietaryguidelines.gov/resources/2020-2025-dietary-guidelines-online-materials/food-sources-select-nutrients/food-1 (accessed on 6 June 2024).

**Table 2 nutrients-16-03350-t002:** Common over-the-counter oral iron formulations.

		Different Preparations	mg Elemental/mg Common Doses	Gastrointestinal Side Effects	Notes
Ferrous salts (Bivalent)	Ferrous sulfate	Tablet, capsule, suspension, extended-release preparation, enteric/film-coated	65/325 tablet	~33%	Ferrous salt is 3–4 times more bioavailable than ferric salts (10–15% bioavailability) Gastrointestinal side effects: constipation, nausea, heartburn, constipation and diarrhea (mostly constipation)
	Ferrous fumarate	Tablet/chewable, in combination with docusate sodium, extended release/film-coated	106/324	~47%	
	Ferrous gluconate	Tablet	38/324	~31%	
Ferric salts (trivalent)	Iron protein succinylate	Liquid	18/300	~7%	3–4 times less bioavailability than ferrous salts. May have better tolerability than ferrous salts. May not be used in patients with milk protein hypersensitivity
	Polymaltose iron complex	Capsule, liquid, film-coated tablet	150 mg/capsule	No data on advantage of tolerance over ferrous sulfate	Poor bioavailability, needs higher-dose intakes, less effective than ferrous sulfate, more expensive
Carbonyl iron		Suspension, tablet, chewable	45 mg/tablet	No clear data on side effect/tolerance benefit over other oral iron preparations	An elemental iron. No evidence of benefit over other iron preparations
Heme Iron Polypeptide		Tablets derived from myoglobin/hemoglobin of animal sources	Variety of products. Example: 65 and 120 mg/tablet	No significant advantages over ferrous sulfate	Higher absorption but more expensive. No superiority to ferrous sulfate in improving hematinic levels

**Table 3 nutrients-16-03350-t003:** Intravenous (IV) iron preparations.

Formulation Name/Abbreviation (Brand Name ^® (Registered Trade Mark)^)		High-Molecular Weight Dextran	Older Generation IV Iron	Older Generation IV Iron Higher Percentage of Labile Iron		Newer Generation IV Iron Less Percentage of Labile Iron Release		
			Low-Molecular-Weight Dextran/LMWD (INFeD^®^)	ferric Gluconate/FG (Ferrlecit^®^)	Iron Sucrose/IS (Venofer^®^)	Ferric Derisomaltose, Iron Isomaltoside/FDI (Monofer/Monoferric^®^)	Ferric Carboxymaltose-/FCM (Injectafer/Ferinject^®^)	Ferumoxytol-FER (Feraheme^®^)
Notes		(OFF THE MARKET)	Needs a test dose. Can be given in one dose	Multiple doses needed. Long infusion time. Test dose is recommended if patient has a history of multiple allergies	Multiple doses needed. Long infusion time. Test dose is recommended if patient has a history of multiple allergies. Non-dextran-derived	Can be given in one dose. Dextran-derived	Can be given in one dose. Non-dextran-derived	Can be given in one dose. Dextran-derived
Iron content			50 mg/mL	12.5 mg/mL	20 mg/mL	100 mg/mL	50 mg/mL	30 mg/mL
Recommended single-dosage administration		(OFF THE MARKET)	1000 mg	125 mg	200–300 mg	1000–2000 mg	750–1000 mg	510–1020 mg
Infusion time for 1000 mg		(OFF THE MARKET)	90–150 min	720 min	300 min	>15 min	>15 min	>15 min
Risks of IV Iron	Major reaction Overall ~1/200,000	Off the market due to high risk of anaphylactic reactions	68/100,000 Black box: risk of anaphylactic reaction—Needs a test dose	24/100,000	24/100,000	Severe hypersensitivity: 0.8%	Severe hypersensitivity: 1.7%	24/100,000
	Fishbane Reaction Mild reaction		~1/200	~1/200	~1/200	~1/200	~1/200	~1/200
	Hypophosphatemia (Unique to newer generations)					3.5%	Higher risk of Hypophosphatemia: ~40% Requires monitoring	0.4%

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
