# Peer review of "Iron Treatment in Patients with Iron Deficiency Before and After Metabolic and Bariatric Surgery: A Narrative Review"

_nutrients, 2024, doi:10.3390/nu16193350_

Round 1
Reviewer 1 Report
Comments and Suggestions for Authors
Undoubtedly, this is a very well-written script. The content of the review has great clinical value and deserves to be published.
A few minor considerations will further enhance the review, make it more appealing and understandable to a wider audience.
1. Line 18- Nevertheless, an evidence-based universal guideline for when and how to treat ID is not yet available.
Can authors please clarify that this is in context of a specific group of patients after MBS. That’s because there do exist guidelines on ID management (ferrous sulphate, gluconate etc. ..). The next sentence on line 19 does indicate this.
2. lines 48, 49,50- However, some studies have suggested effects may be irreversible when anemia is evident despite a hematologic response to treatment
Can the authors please clarify what this means? What is meant by evident anaemia, hematologic response and what treatment are these referring to?
3. line 54- what type of Iron therapy is referred here? intravenous iron infusion, oral?
4. Line 57 – Currently, there is a lack of universal evidence-based guidelines addressing the type, effective dosage, and frequency of iron supplementation to prevent and treat ID and iron deficiency anemia (IDA)
Again, like the Abstract, can the authors please re clarify here that the above sentence refers to a specific group of patients i.e. post MBS patients and not the general population (without MBS) diagnosed with ID. I see that the authors mention this in the remaining part of the above sentence but not stating it here can be misleading.
5. Lines 188 and 189 - In vitro studies have shown that enterocyte uptake of SI is via a vesicle-like structure and is independent of the hepcidin pathway [72]
Can the authors explain in context what they mean by- independent of the hepcidin pathway. Iron sensing mechanisms do not work? or hepcidin is not increased in response to this iron? or any impact on the hepcidin-ferroportin interaction?
Reviewer 2 Report
Comments and Suggestions for Authors
This is a review article focused on the problem of iron deficiency and iron supplementation in obese patients before and after bariatric surgery. The topic is of interest and a lot of data are presented in the paper, however, there are also some important concerns to be addressed.
1. Line 43: “cytochrome” and “peroxidase” are not the single enzymes but a group of enzymes. The text should be modified accordingly or specific cytochromes and specific peroxidases should be listed.
2. What is the prevalence of iron deficiency in obese patients and patients after bariatric surgery vs. non-obese general population?
3. The pathogenesis of ID associated with obesity should be discussed in more details. Authors refer only to chronic low-grade inflammatory state but do not explain its mechanisms and consequences.
4. The other important issues to be discussed are the most commonly used methods of bariatric surgery and the mechanisms through which these procedures may affect iron metabolism. In addition, it is of interest if the risk and the severity of ID depends on the method of bariatric surgery used.
5. The manuscript is a little bit unbalanced, a lot of attention is paid to general characteristics of various iron preparations not to specific aspects of ID associated with obesity or bariatric surgery.
6. Lines 218-219, if formula used to calculate the dose of intravenous iron is mentioned it should be presented in the text.
7. Are there any differences/specificities regarding complications of iv iron administration in the patients after bariatric surgery?
